# Nuclear Receptors and Development of Marine Invertebrates

**DOI:** 10.3390/genes12010083

**Published:** 2021-01-11

**Authors:** Angelica Miglioli, Laura Canesi, Isa D. L. Gomes, Michael Schubert, Rémi Dumollard

**Affiliations:** 1Laboratoire de Biologie du Développement de Villefranche-sur-Mer (LBDV), Institut de la Mer de Villefranche, Sorbonne Université, CNRS, 181 Chemin du Lazaret, 06230 Villefranche-sur-Mer, France; miglioli@obs-vlfr.fr (A.M.); igomes@obs-vlfr.fr (I.D.L.G.); michael.schubert@obs-vlfr.fr (M.S.); 2Dipartimento di Scienze della Terra, dell’Ambiente e della Vita (DISTAV), Università degli Studi di Genova, Corso Europa 26, 16132 Genova, Italy; laura.canesi@unige.it

**Keywords:** nuclear receptors, development, marine invertebrates, evolutionary developmental biology

## Abstract

Nuclear Receptors (NRs) are a superfamily of transcription factors specific to metazoans that have the unique ability to directly translate the message of a signaling molecule into a transcriptional response. In vertebrates, NRs are pivotal players in countless processes of both embryonic and adult physiology, with embryonic development being one of the most dynamic periods of NR activity. Accumulating evidence suggests that NR signaling is also a major regulator of development in marine invertebrates, although ligands and transactivation dynamics are not necessarily conserved with respect to vertebrates. The explosion of genome sequencing projects and the interpretation of the resulting data in a phylogenetic context allowed significant progress toward an understanding of NR superfamily evolution, both in terms of molecular activities and developmental functions. In this context, marine invertebrates have been crucial for characterizing the ancestral states of NR-ligand interactions, further strengthening the importance of these organisms in the field of evolutionary developmental biology.

## 1. Introduction

Nuclear receptors (NRs) are a superfamily of phylogenetically related transcriptional regulators and act as activators or repressors of gene transcription, either constitutively or depending on the binding of a ligand [1,2]. NRs are characterized by a conserved structural organization, comprising a variable N-terminal region (A/B domain), a DNA-binding domain (DBD, C domain), a hinge region (D domain), a ligand-binding domain (LBD, E domain) and a C-terminal domain (Figure 1) [3].

The LBD permits ligand binding, mostly through Van der Waals interactions and hydrogen bonds of specific amino acid residues in the ligand-binding pocket (LBP) [4]. The DBD is the region of the NR that mediates interactions with DNA and hence controls the specificity of the transcriptional response. NRs bind DNA at so-called response elements (REs), specific sequences located in the *cis*-regulatory regions of target genes [5]. REs consist of direct, inverted or palindromic repetitions of a core consensus motif separated by a variable number of nucleotides that are bound by NRs as monomers, homodimers or heterodimers [6]. Efficient dimeric complexes are formed in vivo through cooperative protein-protein and protein-DNA interactions [7]. While protein-protein dimerization interfaces are found in the DBD, the LBD and the hinge region of NRs, REs serve as NR dimerization sites on DNA [8,9,10].

NRs are divided into two main classes: NRs with known ligands and NRs, for which ligands do not exist or have yet to be identified, the so-called orphan receptors [11,12]. Recent phylogenetic analyses of DBD and LBD sequences defined nine NR subfamilies (Table 1), which have probably originated from a single ancestral NR [2,12,13,14,15,16].

NR genes have been found in all extant metazoan taxa, but not in fungi, plants or unicellular eukaryotes, suggesting that NRs originated at the base of the metazoans [1,2]. NR phylogenies have indicated that the ancestral receptor might have been a ligand-activated receptor, with fatty acids as possible ligands [15,16]. In the course of evolution, the NR superfamily has experienced a complex pattern of gene expansions that led to the current diversity of metazoan NRs (Figure 2) [17,18,19,20,21,22,23,24,27].

Genomes of placozoans contain members of NR subfamilies 2 and 3, suggesting that a first wave of NR gene expansion occurred relatively early in metazoan evolution [20,21,29,30]. The NR complement of cnidarians indicates that the diversification of the NR2 and the appearance of the NR6 and NR7/8 subfamilies predate the split of the cnidarian and bilaterian lineages (Figure 2) [20,27,30]. Another NR expansion occurred very early in bilaterian evolution, leading to the appearance of all extant NR subfamilies and an estimated complement of about 25 NR genes (Figure 2) [1,15,27]. The NR superfamily subsequently experienced lineage-specific expansions. The NR3C receptors, for example, arose at the base of the chordates and whole genome duplications further expanded the NR complement in the vertebrate lineage (Table 1, Figure 3) [31,32,33]. A specific lineage-specific duplication has further been reported in cephalochordates, whose genomes encode 10 NR1H receptors (Figure 3) [14]. Similarly, in nematodes, the orphan NR HNF4 experienced a lineage-specific burst of duplications (Figure 3) [34].

Gene losses have also been important for shaping the NR complements of extant metazoans. Ecdysozoans, for example, have lost a whole suite of NRs, including ER, THR, PPAR and NR7/8. RAR was further lost by arthropods and nematodes, and ERR by nematodes only (Figure 3) [24,27,35]. Similarly, within the chordates, NR0A genes have been lost during early diversification of the lineage, tunicates and vertebrates have lost NR7/8, and the tunicates have been subjected to additional NR losses, including, for example, ER, TLL/TLX and NR3C (Figure 3) [14,24,27,31,36].

Despite their presence in all metazoans, NR functions and signaling pathways have primarily been characterized in vertebrates where they were shown to be involved in a wide variety of biological processes, one of which is the regulation of embryonic and post-embryonic development [2,37]. NR names and classifications into orphan or ligand-activated receptors thus strictly reflect their functions and ligand binding properties in vertebrates, with limited relevance to other animal taxa [11,16,25]. In vertebrates, ligand-activated NRs are mainly found in the NR1 and NR3 subfamilies, and their endogenous ligands establish a diverse list of bioactive compounds (Table 1) [25,26]. Furthermore, NR1 and NR4 receptors generally exert their biological functions in heterodimeric complexes with RXR [38,39]. RXR is an orphan nuclear receptor of the NR2 subfamily, whose main biological function is to act as a permissive heterodimeric binding partner of NR1 and NR4 subfamily members, in a process commonly referred to as RXR subordination [39]. Conversely, NR2, NR3, NR5 and NR6 subfamily members mostly function in homodimeric complexes [6].

Given their ligand-dependent activity and their involvement in life threatening human pathologies, such as diabetes and cancer, NRs are major pharmacological targets in several different drug discovery programs [40]. Unfortunately, their ligand-dependent activity also makes them susceptible to a class of environmental pollutants defined as endocrine disrupting chemicals (EDCs), exogenous substances that alter the function of the endocrine system [41,42]. EDCs mimic the structure of endogenous NR ligands and can act as either agonists or antagonists of NR signaling pathways [42,43]. EDC exposure thus adversely affects the adult endocrine system and impacts developmental processes, leading to pathophysiological and pathological conditions, such as neurodevelopmental disorders and developmental dysfunction/arrest in all vertebrate taxa [42,43]. It was initially thought that invertebrates are not affected by EDCs as they were said to lack an endocrine system similar to that of vertebrates [44,45]. However, it is now known that invertebrates are extremely sensitive to EDC exposure, in particular during their embryonic and post-embryonic development [44,45,46]. For this reason, a specific set of NRs, including thyroid hormone (THR), retinoic acid (RAR), retinoid X (RXR) and estrogen (ER, ERR) receptors, is starting to be characterized in invertebrates in an effort to understand their functional traits in relation to those of their vertebrate orthologs. Even though it is still not clear to what extent the teratogenic effects of EDCs in invertebrates are mediated by NRs, it has been shown that NRs are involved in embryonic and post-embryonic development in these animals [47,48,49,50,51,52,53]. Yet, the mechanisms at play are not necessarily conserved with vertebrates, revealing lineage-specific adaptions in both invertebrates and vertebrates [1,10,17,47,48,49,50,51,52,53,54,55,56,57]. The aim of this review is to describe and discuss the developmental functions of NRs in invertebrates, with a special focus on marine organisms, highlighting the particular importance of comparative approaches using emerging marine invertebrate models for the field of evolutionary developmental biology.

## 2. Development of Marine Invertebrates and Associated NR Cohorts

In this section, we will correlate NR expression with the development of marine invertebrates, focusing on representatives from three phyla: cnidarians, mollusks and chordates. Marine invertebrate development can generally be classified into four stages: embryonic development, embryo to larva transition (EtL), larval development and metamorphosis. Embryonic development is characterized by cell proliferation and germ layer specification [47]. EtL consists of a series of morphogenetic processes resulting in the formation of a primitive larva. These processes include axial and body patterning, initiation of neurogenesis and organogenesis [58]. During larval development, additional structures for feeding, light sensing and swimming/crawling are formed [59]. Most of these structures will subsequently be lost during metamorphosis, which results in the emergence of the adult body plan [60]. This typical life cycle can be completed by an asexual reproduction phase. Juvenile polyps of Medusozoan cnidarians, for example, can undergo strobilation, leading to segmentation of the polyp into so-called ephyras, which will subsequently grow into adults [61]. Furthermore, several tunicate lineages independently gained the capacity for asexual reproduction using different budding strategies [62,63].

As detailed above, NR complements can vary greatly between different marine invertebrate phyla. Cnidarians, for instance, possess only a limited number of NRs [20,30,61,64]. The genome of the cnidarian *Nematostella vectensis*, an anthozoan, was thus estimated to encode 17 NRs, including orthologs of vertebrate COUP-TF, TLX/PNR, HNF4, TR2/4 and GCNF [30]. The genomes of medusozoan cnidarians, such as *Aurelia aurita*, include additional NRs, such as orthologs of vertebrate RXR and an ER-like NR, called NR3E (Figure 2; Figure 3) (Table 1) [20,64,65]. In comparison, mollusks have larger NR complements with representatives of each NR subfamily. In the genomes of *Crassostrea gigas*, *Biomphalaria glabrata* and *Lottia gigantea*, for example, 43, 39 and 33 genes encoding NRs were respectively identified, with most of them being orthologous to vertebrate NRs [19,23,24]. While the stereotypical NR complement of chordates is similar to that of mollusks, tunicates are characterized by significantly reduced NR complements. The ascidian tunicate *Ciona intestinalis*, for example, only has 17 NR genes [36]. 

It is intriguing to speculate that differences in NR complements are correlated with the diversity of life cycles, developmental strategies and reproductive adaptations observed in marine invertebrates [66]. NRs are essential regulators of vertebrate development [67,68]. Given that a certain number of developmental processes are conserved in metazoans (or at least bilaterians), it is conceivable that NRs are also pivotal regulators of embryonic and post-embryonic development in invertebrates [48,69]. However, developmental expression of NRs has only been established in a limited number of marine invertebrates, and their developmental functions in marine invertebrates are chiefly unknown [30,47,48,53,65].

In *N. vectensis* (Figure 4), 13 of the 17 NRs are dynamically expressed during development, with distinctive temporal patterns during embryogenesis, EtL, larval development and metamorphosis. While homologs of COUP-TF and TLX/PNR are highly expressed during EtL, HNF4, TR2/4, COUP-TF and NR7/8 expression is characteristic for larval development. Intriguingly, although GCNF is detectable throughout development, its expression increases during late development and marks the metamorphic stage [30]. Furthermore, in *A. aurita*, RXR expression peaks during strobilation (Figure 4) [64,65]. In the bivalve mollusk *C. gigas*, 34 of the 43 NRs are characterized by dynamic expression patterns during development (Figure 4). Orthologs of several NRs, including 2DBD-NRγ, HNF4 and NR7/8, are strongly expressed during embryonic development, with transcripts being almost exclusively of maternal origin [24,47]. While EtL is characterized by a number of NR transcripts including THR, RAR, PPAR, RXR, Rev-ErbA, TLL/TLX and PNR, larval development is marked by a general downregulation of NR expression [24,47,70]. Metamorphosis, in turn, is characterized by NR0B, 2DBD-NRδ, EcR, RXR, COUP-TF/SVP, TLL/TLX, ER and ERR expression [47]. Intriguingly, various members of the *C. gigas*-specific NR1P subgroup are dynamically expressed during development [17,47]. In the ascidian tunicates *C. intestinalis* and *Phallusia mammillata*, a principal component analysis of NR transcripts has not yet been performed. However, the expression profiles of ascidian NRs available in the Aniseed database (https://www.aniseed.cnrs.fr/) allow at least some level of developmental clustering (Figure 4). Embryonic development is thus characterized by expression of TR2/4, RXR, PPAR and LXR, all of which are of maternal origin. EtL is associated with RAR, HNF4, GCNF and ROR expression, while peaks in the expression of THR, Rev-Erb, ERR, COUP-TF and PPAR occur during larval development [53]. Altogether, these results demonstrate that the development of marine invertebrates is characterized by the stage-specific expression of NR subsets, suggesting that NR activity is correlated with distinctive developmental processes at different stages of the life cycle.

## 3. Development of Marine Invertebrates and NR Diversification 

Despite the gene duplications and losses that accompanied NR diversification, a basic set of NRs is implicated in the development of distantly related metazoans. This basic set of NRs comprises COUP-TF/SVP, TLX/PNR, HNF4 and RXR (Figure 5, Table 1). Information regarding their developmental functions in marine invertebrates is still extremely scarce and is largely derived from the biological activity of their vertebrate orthologs.

### 3.1. Chicken Ovalbumin Upstream Promoter Transcription Factor, COUP-TF

In vertebrates, COUP-TF generally acts as a transcriptional repressor and regulates the development of muscles and heart as well as the differentiation of hindbrain and photoreceptors [71,72]. COUP-TF orthologs are expressed in early and late larval stages of *N. vectensis* and in late pre-metamorphic stages of both *C. gigas* and *C. intestinalis* (Figure 4). As this receptor is expressed in nematoblasts and subsets of neural cells in the cnidarian *Hydra vulgaris* as well as in the posterior photoreceptive ocellus of *P. mammillata*, COUP-TF is considered as a conserved neural marker involved in the formation of photoreceptive organs [53,73].

### 3.2. Tailless/Photoreceptor Cell-Specific Nuclear Receptor, TLX/PNR 

In vertebrates and fruit flies, TLL/TLX is an orphan receptor involved in eye and forebrain development as well as in anteroposterior patterning of the embryo, suggesting some level of functional conservation between vertebrates and invertebrates [48,74]. In both *N. vectensis* and *C. gigas*, TLX/PNR orthologs are highly expressed in EtL stages (Figure 4). Their predominant expression in neural tissues is indicative of functions in neurogenesis and in the development of photoreceptive organs [30,47]. While TLX/PNR receptors were secondarily lost in tunicates, the TLL/TLX ortholog of the cephalochordate amphioxus is dynamically expressed in EtL stages. In neurulae, the amphioxus TLL/TLX gene marks developing sensory neurons, and in larvae, the gene is expressed in the central nervous system and anterior notochord [75]. It is thus likely that TLX/PNR genes are also involved in neurogenic processes in developing invertebrate chordates.

### 3.3. Hepatocyte Nuclear Factor 4, HNF4

In humans, HNF4 binds endogenous fatty acids as ligands and regulates hepatocyte differentiation, energy metabolism, xenobiotic detoxification and stem cell maintenance in the germ line [76,77]. HNF4 also participates in primary endoderm development in frogs, regulates expression of transcription factors necessary for endoderm specification in mice and is required for gut formation in insects [78,79,80]. HNF4 is highly expressed during larval development in *N. vectensis*, is a maternal transcript in *C. gigas* early embryos and expressed at EtL stages in *C. intestinalis* (Figure 4) [30,47]. In *P. mammillata,* HNF4 is expressed in endoderm cells of the trunk [53]. Developmental expression of HNF4 could thus play a role in endoderm specification and the formation of endodermal organs [30,47,53].

### 3.4. Retinoid X Receptor, RXR 

In vertebrates, RXR is commonly the silent heterodimeric partner of NR1 and NR4 subfamily members and is involved in a variety of developmental processes in subordination to its heterodimeric binding partners. In the medusozoan cnidarian *A. aurita*, RXR is dynamically expressed during early strobilation, suggesting a potential role for RXR in this asexual reproduction process (Figure 4) [65]. RXR is highly expressed in *C. gigas* EtL stages, together with the NR1 subfamily members THR, RAR and PPAR [47,70]. In addition, there is a second peak of RXR expression prior to metamorphosis, which is paralleled by the NR1 subfamily member EcR [47]. Conversely, RXR in *C. Intestinalis* is expressed mainly during embryonic development, together with PPAR, and less strongly during EtL and larval stages, which are, respectively, characterized by the expression of the NR1 subfamily members RAR and ROR and THR, Rev-erb and PPAR [53]. While the developmental clustering of RXR does not strictly follow that of the NR1 subfamily members, there is nonetheless a tendency for regrouping RXR expression with that of the representatives of the NR1 subfamily. The notable absence of NR4 subfamily members from the developmental clusters in cnidarians, mollusk and tunicates suggests that heterodimers of RXR and NR1 receptors might play much more important roles during invertebrate development than heterodimers of RXR and NR4 receptors.

The receptors defining the basic set of NRs acting during development of marine invertebrates are all members of the NR2 subfamily. This subfamily has appeared very early in the metazoan lineage and has diversified before the cnidarian-bilaterian split [27,30]. Of all NRs, the NR2 subfamily member HNF4 is actually considered the extant NR that most closely resembles the ancestral NR that originated in the last common ancestor of all metazoans [15,16]. Based on the observations detailed above, it thus seems likely that the early diversification of the NR2 subfamily was accompanied by the elaboration of a NR2-dependent gene regulatory network involved in different aspects of animal development [81]. If this hypothesis is correct, at least some elements of this core gene regulatory network should still exist in extant animals and control conserved developmental functions in different metazoans. The basic set of NRs identified here and their potential involvement in invertebrate development could serve as a starting point for future studies aimed at identifying these ancestral NR-dependent features of animal development.

## 4. Thyroid Hormone Receptor (THR) Signaling Regulates Developmental Transitions in Marine Invertebrates

In vertebrates, THR is a ligand-activated transcription factor and its ligands are generally referred to as thyroid hormones (THs), which are either synthesized endogenously or taken up from the environment (Table 1) [82,83]. The main THs of vertebrates are triiodothyronine (T3) and tetraiodothyronine (T4), with T3 being the biologically active TH. THs are key regulators of vertebrate development and homeostasis and are involved, for example, in animal growth and metabolism as well as in the regulation of metamorphosis [84]. THR is dynamically expressed during development of both mollusks and tunicates, which is suggestive of a possibly conserved function in bilaterian development (Figure 5). THR genes originated at the base of bilaterians and have already been identified in the genomes of a wide variety of protostomes and deuterostomes [10,52,85]. However, the THR gene was lost in ecdysozoans and lineage-specific THR duplications occurred, for example, in platyhelminths, likely by independent duplication events in trematodes and turbellarians (Figure 3) (Table 2) [27,85].

Ascidian tunicates possess a THR ortholog and can endogenously produce T4 [86,87,88]. T4 is present in pre-metamorphic stages, and experimental evidence suggests that it could be a regulator of metamorphosis [87,88]. However, a direct involvement of THR in this process remains elusive [86,87]. In the cephalochordate amphioxus, several elements of the THR signaling system are shared with vertebrates, but the biologically active TH is triiodothyroacetic acid (TRIAC), rather than T3 [89,90]. TRIAC binds and strongly activates the amphioxus THR, and TH-dependent signaling plays a pivotal role in the regulation of metamorphosis [89,90]. In echinoderms, a THR gene has been cloned from sea urchins, but it has been shown that this receptor is not activated by THs or their metabolites [52,83]. Yet, T3 and T4 are actively accumulated during echinoderm development, and an exogenous supply of T4 can accelerate development, skeletogenesis and metamorphosis in a variety of echinoderm species [52,91].

Ligand-controlled THR signaling has further been suggested to regulate development and metamorphosis in different protostomes. However, in the absence of convincing evidence of endogenous TH synthesis in protostomes, it is currently believed that protostomes have to take up THs, or its precursors, from external sources [52,54,92]. In annelids, the THR of *Platynereis dumerilii* is activated in the presence of T3 or TRIAC [10,54]. Treatments with exogenous T3 or TRIAC induce an acceleration of the morphological switch from the trochophore to the crawling larva. This morphological switch is further characterized by a peak of THR expression [10,54]. The THR of the annelid *P. dumerilii* might thus be a ligand-activated receptor with endogenous ligands similar to T3 or TRIAC that mediates developmental transitions between larval stages [54,93].

In mollusks, experimental evidence also points to a morphogenetic function of THR and THs. While the transcriptional activity of *C. gigas* THR is not stimulated at relevant physiological concentrations of T4, T3 or TRIAC in vitro, both T4 and T3 are present in vivo in embryos and larvae, with their concentrations increasing significantly between the gastrula and the feeding larva [94]. In addition, the THR protein is detectable from blastula to trochophore stages in *C. gigas*, suggesting that the THR and the TH signaling system might be involved in the regulation of the embryo to larva transition [94]. Similarly, although platyhelminth THRs are not activated by THs, exogenous THs accelerate the development of parasitic platyhelminth lineages [85,95,96]. Taken together, it seems likely that THs have an ancestral role in lophotrochozoan development and that THRs are involved in mediating these roles. However, the endogenous ligands of lophotrochozoans THRs seem to be different from those of vertebrate THRs.

THs were further shown to function in developmental transitions in animals that lack a THR, such as cnidarians where THs regulate metamorphosis and strobilation [52,83]. A role for TH signaling in the control of developmental transitions might thus predate the evolutionary origin of THRs [52,54,83]. It has been proposed that the first TH signaling systems to evolve used TH precursors obtained from algal sources and that the ancestral THR evolved as a sensor for iodinated tyrosine and indicator of food availability [10,52,83,92]. The appearance of THR in bilaterian animals thus allowed for the elaboration of a ligand-dependent control mechanism for development and growth, whose activity is directly coupled to environmental cues favoring larval survival [10,52,83,92,97].

## 5. Retinoic Acid Receptor (RAR)-Dependent Signaling Is Required for Neurogenesis in Marine Invertebrates

As in the case of THR, RAR is also dynamically expressed during larval morphogenesis in invertebrates (Figure 5). It is thus reasonable to assume that this NR is also involved in the regulation of developmental functions in marine invertebrates. Vertebrate RARs are ligand-activated transcription factors that act as constitutive repressors in the absence of a ligand [98]. RARs bind different isomers of the small, lipophilic molecule retinoic acid (RA), such as all-*trans*-RA, 9-*cis*-RA or 13-*cis*-RA, with all-*trans*-RA being the main biologically active RA isomer [99]. RA acts as a morphogen, whose functions are mediated by RARs. During vertebrate development, RA signaling is, for example, required for axial patterning, nervous system development and organogenesis, with HOX genes being amongst the major targets of this signaling pathway [99,100]. Most bilaterian genomes contain a single RAR gene [17,27,36,49,50,101,102,103]. However, RARs have been lost in most ecdysozoans as well as in appendicularian (larvacean) tunicates (Figure 3) (Table 3) [27,101,103].

Many elements of a vertebrate-like RA signaling system are present in invertebrate deuterostomes, including the genes encoding the receptors and the enzymes required for the synthesis and degradation of endogenous RA [99]. In tunicates that have not secondarily lost the RAR gene, RAR-dependent RA signaling is implicated in neurogenesis as well as in tissue regeneration and bud development of budding tunicates [104,105]. The cephalochordate amphioxus is characterized by the most vertebrate-like RA signaling system of all invertebrates [99,106]. It has been demonstrated that the amphioxus RAR/RXR heterodimer can be activated by all-*trans*-RA, that HOX genes are directly regulated by RAR/RXR and that HOX-mediated RA signaling is essential for neurogenesis and axial patterning [107]. Conversely, little is known about RAR and RA signaling in ambulacrarians. In echinoderms, the sea urchin RAR can bind RA in vitro, although with low affinity, and, while RA treatments might either disrupt or delay sea urchin development, the molecular mechanisms underlying these effects remain to be established [49,99].

In lophotrochozoans, extensive functional characterizations of RAR orthologs and RA signaling have been carried out in both annelids and mollusks [49,50,51,55,102]. The annelid *P. dumerilii* possesses a vertebrate-like RA signaling machinery composed of a ligand-activated RAR as well as enzymes for RA synthesis and degradation [49,101]. *P. dumerilii* RAR is activated by RA binding and regulates transcription in a heterodimer with RXR [49]. However, the *P. dumerilii* RAR has a lower affinity for RA than its vertebrate orthologs, and the conformation of the ligand within the LBP is not conserved between the annelid and vertebrate receptors (Figure 1) [49]. When exposed to all-*trans*-RA or 13-*cis*-RA, *P. dumerilii* embryos experience neuroblast depletion leading to reduced numbers of differentiating motor neurons and suggesting a direct effect of RA on dividing neural stem cells [49]. Moreover, knockdown of *P. dumerilii* RAR or RXR causes severe malformations of the developing larval nervous system [49]. RAR-dependent RA signaling is thus required for neurogenesis in annelids [49].

Contrasting the situation in annelids, mollusk RARs have lost the ability to bind RA. It has been suggested that mollusk RARs secondarily lost the capacity to bind RA by accumulating independent single mutations in different mollusk lineages [51,56]. Accordingly, mollusk RARs likely function as constitutive transcriptional repressors [50,51,102]. It has recently been proposed that this repressive activity of RAR within the RAR/RXR heterodimer could be modified by RXR-dependent ligand binding [50,102]. It thus remains elusive how RA controls development and neurogenesis in mollusks [50,55,56,99,106]. Similarly, although the genomes of most ecdysozoans do not encode a RAR gene, there are indications for active roles of RA signaling in ecdysozoans, for example, during nervous system regeneration and in tissue repair of insects [108,109]. As a matter of fact, RAR has been lost very early during ecdysozoan evolution, after the split of the priapulid lineage [103]. Notably, the RAR of the priapulid *Priapulus caudatus* binds all-*trans*-RA and 9-*cis*-RA with affinities similar to those of the *P. dumerilii* RAR [49,103]. In cnidarians, whose genomes do not encode RARs, RA has been implicated in the regulation of metamorphosis and neural development [55,64,99,110]. Independent of the presence of RAR, RA can thus be considered a potent morphogen involved in metazoan neurogenesis. Future work will have to address how the RA signal is mediated in the absence of RAR, and several scenarios, including the involvement of a liganded RXR, have already been proposed for the evolutionary origin of RA signaling at the base of metazoans [50,99]. RARs only arose later, at the base of bilaterians, as ligand-activated RA sensors that significantly facilitated the regulatory control of RA signals [49].

## 6. Retinoid X Receptor (RXR) Functions during Marine Invertebrate Development

In vertebrates, RXRs act as heterodimeric binding partners of NR1 and NR4 subfamily receptors. Their activity is thus subordinated to those of the heterodimeric binding partners, which participate in a wide variety of developmental processes [39,111]. However, under very specific circumstances, RXRs may also function as ligand-activated receptors. In vitro, RXRs are activated by a number of different compounds, including 9-*cis*-RA and fatty acids, such as docosahexaenoic acids (DHAs) [112,113]. Orthologs of vertebrate RXRs have been identified in most metazoan taxa, including sponges, placozoans and cnidarians, suggesting that RXRs originated at the base of the animal tree of life [57,114]. Lineage-specific duplications of RXR genes are rare and have so far only been reported in two lophotrochozoans: bryozoans and platyhelminths (Figure 3) [57]. RXR is widely expressed during development of most marine invertebrate taxa, including cnidarians, mollusks and tunicates. Compared to other NRs, RXR does not show a clear pattern of developmental clustering. Its expression profiles tend to follow the ones of its heterodimeric binding partners (Figure 4).

Marine invertebrates provide excellent examples for the importance of RXRs as regulators of development and point to a possible ligand-activated function of these receptors (Table 3). In the sea urchin *Strongylocentrotus nudus*, for example, an ambulacrarian deuterostome, knockdown of RXR induces abnormal early embryonic development and leads to a complete arrest of embryonic development at the early gastrula stage, suggesting that RXR is crucially required for sea urchin development [115]. Furthermore, a study of the RXR from the sea urchin *Paracentrotus lividus* suggests that exposure to an endogenous gonadal fatty acid mixture stimulates the activity of the PPAR/RXR heterodimer *in cellulo* and that this stimulation is mediated by ligand binding to RXR. In this particular context, the transcriptional activity of the PPAR/RXR heterodimer might thus be regulated by RXR [116]. In ecdysozoan and lophotrochozoans protostomes, RXR mainly acts as a heterodimeric partner of other NRs, in accordance with the process of RXR subordination [117,118]. However, a ligand-activated function of RXRs has been proposed in mollusks, where it was shown, in *C. gigas*, *Nucella lapillus* and *Acanthochitona crinita*, that RXR-specific ligands abrogate the repressive state of RAR/RXR heterodimers [50]. Given that mollusk RARs do not seem to bind RA, the activity of RAR/RXR during mollusk development might thus be regulated by ligand binding to RXR. Further studies need to address this hypothesis in vivo and evaluate the biological relevance of RXR ligands during development.

While the genomes of non-bilaterians encode RXR genes, their heterodimeric binding partners, i.e., members of the NR1 and NR4 subfamilies, are absent. The RXR of medusozoan cnidarians binds 9-*cis*-RA with high affinity and is required for strobilation [64]. Cnidarian RXRs might further be involved in the RA-dependent regulation of nervous system patterning and neurogenesis [65,99,110]. It is thus tempting to speculate that RXR is mediating the developmental roles of RA in cnidarians, potentially as a homodimer [119].

## 7. Estrogen Receptor (ER), Estrogen-Related Receptor (ERR) and the Development of Marine Invertebrates

In vertebrates, ERs are activated by estradiol (E2) and function during the formation of the nervous system, during development of secondary sexual characteristics, in the regulation of the immune system, in the maintenance of bone density and in the control of social behavior [120,121,122]. ERRs form a group of orphan receptors that are closely related to ERs and that are critical for the regulation of neurogenesis and metabolism as well as for cell proliferation and cell movements [123,124,125]. While the endogenous ligand of ERRs still remains elusive, their activity can be modulated by synthetic ligands, some of which also activate or inhibit ERs, such as, respectively, diethylstilbestrol (DES) or 4-hydroxytamoxifen (4-HT) [126]. ER and ERR are members of the NR3 subfamily, which originated early in metazoan evolution, likely after the split of the sponge lineage, and experienced diversification in early bilaterians and subsequently also in early chordates (Figure 2; Figure 3) [20,21,127]. Phylogenetically, non-chordate ERs thus group at the base of chordate ERs and SRs [20,127,128]. Single orthologs of ER and ERR have been identified in most protostomes, with ecdysozoans having lost ER and the nematodes, which are ecdysozoans, having additionally lost ERR (Figure 3) [27]. The genomes of invertebrate deuterostomes also encode single copies of ER and ERR and those of cephalochordates additionally a single SR [14,31]. Within chordates, the tunicates have secondarily lost both ER and SR (Figure 3) [27,36,127]. The lineage-specific loss of an ER-like gene has further been reported in anthozoan cnidarians (Figure 3) [20,30]. Even though their expression profiles are indicative of a function during invertebrate development (Figure 5), the role played by these NRs outside vertebrates remains virtually unknown.

### 7.1. ERR Might Play a Role during Development of Marine Invertebrates

Ascidian tunicates possess a single ERR gene [36,53], whose expression, albeit dynamically modulated during late larval development, is exclusively localized in the brain (Figure 4) (Table 4) [53,129]. DES and 4-HT both affect brain formation and trunk elongation in developing *P. mammilata* larvae, suggesting that ERR acts during nervous system development and body extension of tunicates [53,129]. In the cephalochordate amphioxus, ERR is expressed in the hindbrain homolog and in the developing musculature [130]. The segmented patterns of ERR expression in hindbrain and muscles suggest that this NR is involved in establishing neuromuscular contacts in developing amphioxus [130]. In protostomes, ERR functions have been established in ecdysozoans, where it regulates metabolic processes supporting larval growth and cell proliferation [131,132]. ERR is further suspected to be a pivotal player during metamorphosis [131]. In contrast, there is only limited information available on ERR expression and functions in lophotrochozoans. In *C. gigas*, ERR is dynamically expressed during development with a peak at pre-metamorphic and metamorphic stages [47]. Accordingly, it is parsimonious to assume that ERR plays specific functional roles during development of mollusks, and more generally, of lophotrochozoans.

### 7.2. Developmental Functions of ER in Marine Invertebrates Remain Largely Elusive

In the cephalochordate amphioxus, estrogens have been proposed to play important roles in reproductive functions, such as spawning [133]. However, it is currently unknown if ER and SR are required to mediate these functions (Table 4). In contrast, experimental evidence from marine protostomes, such as annelids and rotifers, supports the hypothesis that ERs are involved in reproductive processes of at least some invertebrates [134,135,136,137]. However, studies addressing the developmental roles of ER-like receptors in protostomes remain extremely rare, with the only description of developmental functions of ER-like receptors in protostomes coming from annelids. In fact, estrogens are endogenously synthesized in annelids and are required for the formation and proliferation of primordial germ cells [136,137,138]. This regulation is dependent on ER, which binds the endogenous estrogens and hence directly controls this process [136,138]. Estrogen also induces proliferation of primitive germ cells in vertebrates, suggesting that germ cell regulation represents an ancestral trait of ER signaling in bilaterians [138]. In contrast, estrogen binding and ligand-dependent activation were lost in mollusk ERs [128,139]. Mollusk ERs are constitutively active and retained the ability to regulate their own gene transcription, but do not bind estrogens or other steroids, as their LBPs underwent vestigialization [128,139,140]. Nevertheless, in the bivalve mollusks *C. gigas* and *Mytilus galloprovincialis*, dynamic ER expression was detected at larval stages, which is suggestive of a role for ER in larval development [47,141]. Outside bilaterians, ER-like receptors have been identified in placozoans and cnidarians [20,21]. While their expression and function have yet to be assessed, it has been shown that the *H. vulgaris* ER-like receptor can bind paraestrol A, an ancestral estrogen [20,134]. Notably, a compound isolated from the cnidarian *Dendronephthya studeri* is structurally similar to paraestrol A, suggesting that ER-like receptors might act as ligand-activated transcription factors, at least in cnidarians [134,142].

## 8. Are NRs a Primary Target of EDCs in Marine Invertebrates?

Invertebrate protostomes and deuterostomes were initially thought not to be affected by EDCs, as they supposedly lack an elaborate endocrine system [44]. This assumption was shown to be incorrect, following the collapse of gastropod and bivalve mollusk populations in the Arcachon Bay in France. It was later shown that the mollusks were affected by EDCs in marine antifouling paints, already known to modulate the activity of PPAR/RXR heterodimers in vertebrates [45]. Today, it is well established that marine invertebrates are sensitive to EDC pollution and clearly manifest their adverse effects, with embryonic stages being particularly vulnerable. The scientific community has thus invested greatly in the study of EDC exposure and endocrine systems of marine invertebrates, turning their embryos into valuable alternative models for EDC and chemical testing [143]. However, with the exception of the RXR-dependent imposex phenotype in gastropod mollusks [144], evidence for NRs as primary targets of EDCs in marine invertebrates is still extremely circumstantial and mainly based on in vitro studies that do not necessarily reflect the molecular mechanisms in vivo [44,45,46,145,146]. This does not mean that NR-mediated endocrine disruption does not occur on a large scale in marine invertebrates. In fact, the chemicals currently classified as EDCs have chiefly been identified and characterized in vertebrates, and most studies into marine invertebrates have simply used these compounds to assess whether they also affect their endocrine system [46]. This vertebrate-centric view has thus heavily biased our current understanding of invertebrate endocrine systems.

Based on the work reviewed here, it is not surprising that high affinity EDCs of vertebrate NRs do not necessarily affect NRs of marine invertebrates (and vice versa) [44,45,145]. In the course of their evolution and diversification, NRs have been subjected to significant alterations of their sequences and structures, which explains why orthologous NRs are not necessarily characterized by conserved ligand binding affinities, downstream targets and biological functions [1,2,13,15,16,17,25]. Accordingly, there are probably numerous substances that can act as NR-mediated EDCs in marine invertebrates, and their modes of action are likely to be very different from those known in vertebrates [46,146]. Furthermore, it is currently impossible to estimate to what extent the endocrine systems of marine invertebrates are comparable to those of vertebrates, including the involvement of NRs [147,148,149]. There are thus a number of different points that need to be addressed in order to establish the endocrine disrupting potential of a particular set of chemicals in marine invertebrate taxa. These include the characterization of the roles of NRs in marine invertebrate physiology and endocrine systems, the assessment of their functions in embryonic and post-embryonic development and the detailed definition of the physiological and developmental outcomes of NR-dependent endocrine disruption.

## Figures and Tables

**Figure 1 genes-12-00083-f001:**
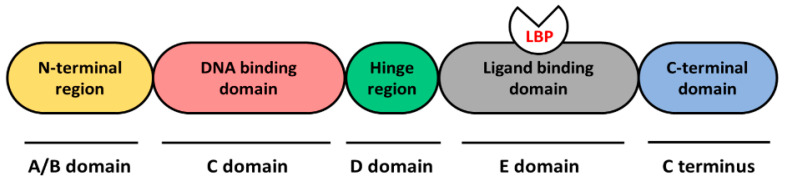
General structure of nuclear receptor (NR) proteins. Different domains of the NR protein are shown in different colors. A schematic representation of the ligand-binding pocket (LBP) is included in the ligand binding domain.

**Figure 2 genes-12-00083-f002:**
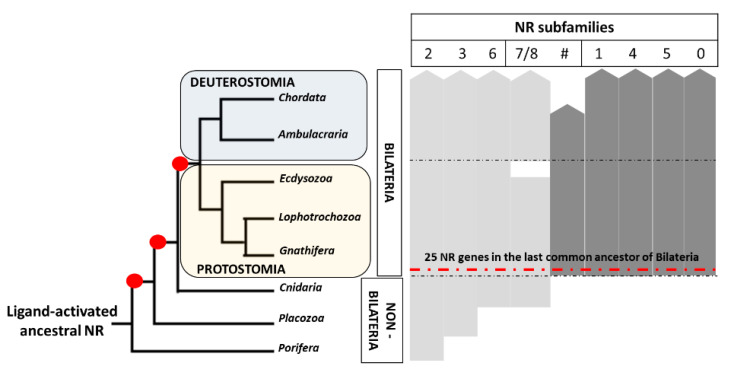
Evolution of the nuclear receptor (NR) superfamily. Subfamily origin and taxonomic distribution in metazoans. Animal phylogeny is based on Laumer et al. (2019) [28]. Subfamily diversification is based on data reported in the text [17,18,19,20,21,22,23,24,27]. Classification of NR subfamilies is as defined in Table 1. Red dots indicate events of subfamily diversification. The dotted red line, on the right, marks the NR expansion event at the base of bilaterians. Black thin dotted lines, on the right, indicate the separation between different animal clades (non-Bilateria/Bilateria and Protostomia/Deuterostomia). Grey bars, on the right, indicate the origin and conservation of NR subfamilies in metazoans. Light grey bars indicate NR subfamilies that were already present before the origin of bilaterians. Dark grey bars highlight bilaterian-specific NR subfamilies.

**Figure 3 genes-12-00083-f003:**
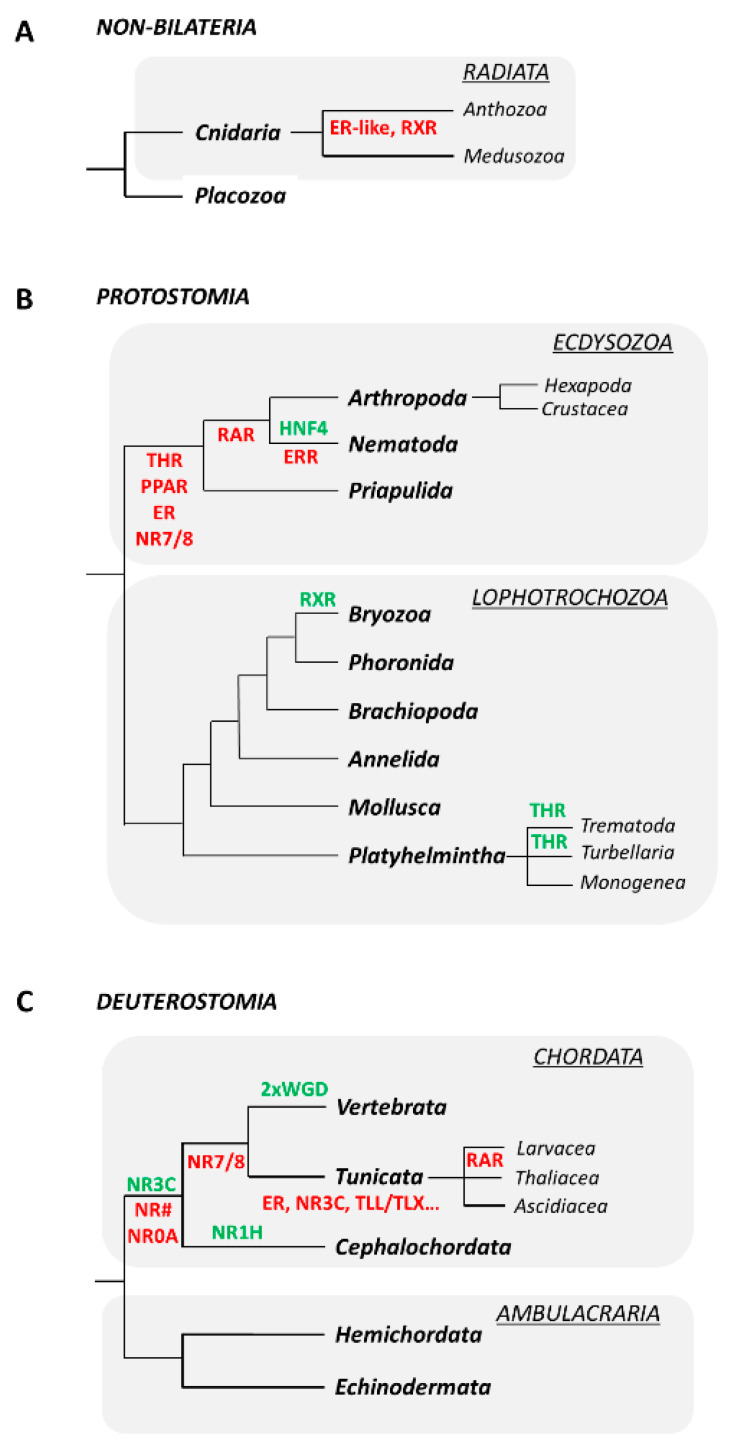
Examples for nuclear receptor (NR) duplications and losses during metazoan evolution. (**A**) Non-bilaterians. (**B**) Protostomes. (**C**) Deuterostomes. Gene expansions are shown in green and gene losses are shown in red. WGD: Whole genome duplications. Animal phylogenies are based on Laumer et al. (2019) [28]. NR nomenclature is as defined in Table 1.

**Figure 4 genes-12-00083-f004:**
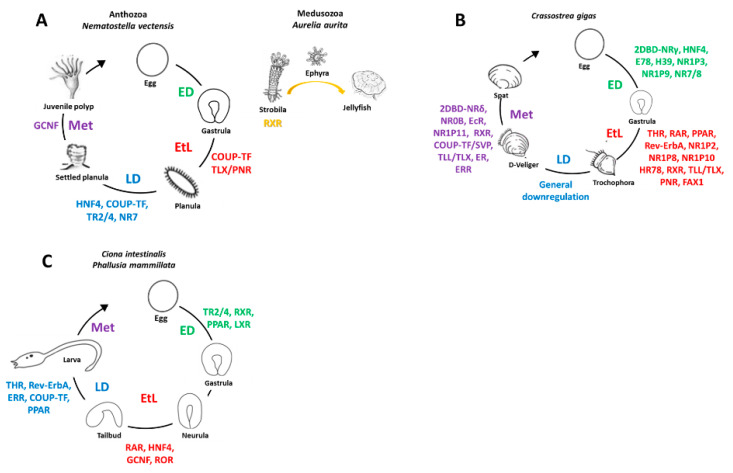
Nuclear receptor (NR) expression during development of marine invertebrates. (**A**) Cnidarians. (**B**) Bivalve mollusks. (**C**) Ascidian tunicates. NRs dynamically expressed during embryonic development (ED) (green), embryo to larva transition (EtL) (red), larval development (LD) (blue), metamorphosis (Met) (violet) and during cnidarian strobilation (orange). NR nomenclature is as defined in Table 1. Data for the cnidarians *Nematostella vectensis* and *Aurelia aurita* from Reitzel et al. (2009), Fuchs et al. (2014) and Brekhman et al. (2015) [30,64,65], for the bivalve mollusk *Crassostrea gigas* from Vogeler et al. (2016) and Huang et al. (2015; 2020) [19,24,70]. Raw data for the ascidian tunicates *Ciona intestinalis* and *Phallusia mammillata* are from the Aniseed database (https://www.aniseed.cnrs.fr/) and from Gomes et al. (2019) [53].

**Figure 5 genes-12-00083-f005:**
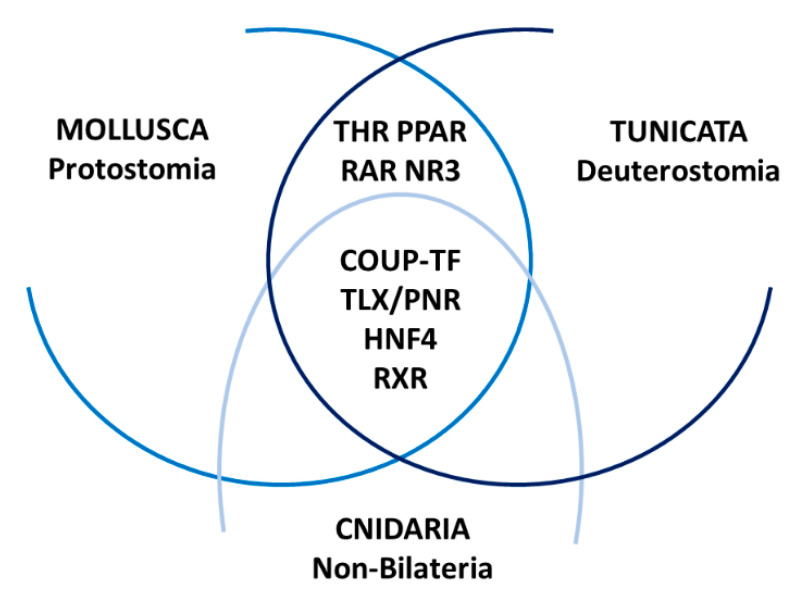
Ancestral sets of nuclear receptors (NRs) define the development of marine invertebrates. Members of the NR2 subfamily (COUP-TF, TLX/PNR, HNF4, RXR) are dynamically expressed during development of cnidarians, mollusks and tunicates. Albeit their presence in the genomes of at least medusozoan cnidarians, NR3 receptors only contribute to the developmental NR complements of the two bilaterian taxa, mollusks and tunicates. Members of the NR1 subfamily (THR, PPAR, RAR) are dynamically expressed during development of both protostomes and deuterostomes, further expanding the bilaterian set of NRs with developmental functions.

**Table 1 genes-12-00083-t001:** Nuclear receptor (NR) complements characterized in metazoan genomes.

Subfamily	Group	Name	NRNC Symbol	Abbreviation	Physiological Ligand
# *	A	NRs with two DBDs	NR#A1	2DBD-NRα	
	B		NR#B1	2DBD-NRβ	
	C		NR#C1	2DBD-NRγ	
	D		NR#D1	2DBD-NRδ	
0	A *	Zygotic gap protein	NR0A1	KNI	x
		Zygotic gap protein- related	NR0A2	KNRL	x
		Egon	NR0A3	EG	x
		ODR-7	NR0A4	ODR-7	x
		Trithorax	NR0A5	TRX	x
	B	Dosage-sensitive sex reversal-adrenal hypoplasia congenital critical region on the X chromosome, gene 1	NR0B1	DAX1	x
		Small heterodimer partner	NR0B2	SHP	x
1	A	Thyroid hormone receptor	NR1A1,2	THRα,β	T3
	B	Retinoic acid receptor	NR1B1-3	RARα-γ	All-*trans*-RA
	C	Peroxisome proliferator-activated receptor	NR1C1-3	PPARα-γ	Fatty acids, Prostaglandins
	D	Rev-ErbA	NR1D1,2	Rev-ErbAα,β	x
	E *	Ecdysone-regulated E78 gene	NR1E	E78	
	F	RAR-related orphan receptor	NR1F1-3	RORα-γ	x
		HR3 *	NR1F4	HR3	
	G *	CNR14-like	NR1G1	Sex-1	x
	H	Liver X receptor-like	NR1H1 *	EcR	Ecdysteroids
			NR1H2,3	LXRα,β	Oxysterols
			NR1H4,5	FXRα,β	x
	I	Vitamin D receptor-like	NR1I1	VDR	Vitamin D
			NR1I2	PXR	Xenobiotics
			NR1I3	CAR	Androstane
	J *	NHR96	NR1J1	DHR96	
	K *	VDR/PXRα,β	NR1K1,2	VDRα,β-like	
	L *	HNR-like 97	NR1L	HR97	
	M *	HNR-like 19	NR1M1	HR10	
	N *	HNR-like 11	NR1N1	HR11	
	O *		NR1O		
	P *		NR1P1-11		
2	A	Hepatocyte nuclear factor 4	NR2A1-3	HNF4α,γ	Fatty acids
			NR2A4 *	HNF4	
	B	Retinoid X receptor	NR2B1-3	RXRα-γ	x
			NR2B4	USP	x
	C	Testicular receptor	NR2C1	TR2	x
			NR2C2	TR4	x
	D *	DHR78	NR2D1	HR78	
	E	Tailless / Photoreceptor cell-specific nuclear receptor	NR2E1	TLX	x
			NR2E2 *	TLL	x
			NR2E3	PNR/HR51 *	x
		Dissatisfaction nuclear receptor *	NR2E4	DSF	
		Nuclear hormone receptor FAX-1 *	NR2E5	FAX1	
	F	Chicken ovalbumin upstream promoter transcription factor	NR2F1,2	COUP-TFI,II	x
		Seven-up *	NR2F3	SVP	
		Chicken ovalbumin upstream promoter transcription factor III *	NR2F4	COUP-TFIII	
		Seven-up related protein 46 *	NR2F5	SVP-46	
		V-erbA-related protein 2	NR2F6	EAR-2	x
3	A	Estrogen receptor	NR3A1,2	ERα,β	Estradiol
	B	Estrogen-related receptor	NR3B1-3	ERRα-γ	x
			NR3B4 *	ERR	x
	C	Steroid receptor / Ketosteroid receptors	NR3C1	Glucocorticoid receptor, GR	Cortisol
			NR3C2	Mineralocorticoid receptor, MR	Aldosterone
			NR3C3	Progesterone receptor, PR	Progesterone
			NR3C4	Androgen receptor, AR	Testosterone
	D *	Estrogen receptor-like in Protostomia	NR3D	ER-like	
	E *	Estrogen receptor-like in Cnidaria	NR3E	ER-like	
	F *	Estrogen receptor-like in Placozoa	NR3F	ER-like	
4	A	Nerve growth factor IB	NR4A1	NGFIB	x
		Nuclear receptor related 1	NR4A2	NURR1	x
		Neuron-derived orphan receptor 1	NR4A3	NOR1	x
		DHR38 *	NR4A4	HR38	
5	A	Steroidogenic factor 1	NR5A1	SF1	Phosphatidylinositols
		Liver receptor homolog-1	NR5A2	LRH1	Phosphatidylinositols
		NHR FTZ1-α *	NR5A3	FTZ1-α	
	B *	NHR39/FTZ1-β	NR5B1	HR39	
6	A	Germ cell nuclear factor	NR6A1	GCNF	x
		HR4 *	NR6A2	HR4	
7/8 *	A		NR7/8A1		

Non-vertebrate NR subfamilies and receptors are indicated by (*). Physiological ligands refer to in vivo conditions in vertebrates and the (x) highlights orphan NRs [12,13,14,15,16,17,18,19,20,21,22,23,24,25,26]. NRs with two DBDs are listed as (#), because they do not yet have an unequivocal subfamily status. Like NR0s, NRs with two DBDs do not have the conventional NR structure as shown in Figure 1. The NR7/8 subfamily has been called either NR7 or NR8 [14,22,24]. When information is absent or disputed, cells are left empty.

**Table 2 genes-12-00083-t002:** Summary of thyroid hormone receptor (THR) functions in marine invertebrates.

Taxon	Clade	Phylum	Receptor Activity	Developmental Function
*Deuterostomia*	*Chordata*	*Tunicata*	Unknown	Suspected role in metamorphosis
		*Cephalochordata*	Activated by TRIAC	Pivotal regulator of metamorphosis
	*Ambulacraria*	*Echinodermata*	Presumably ligand-activated and/or controlled by alternative signaling pathways	Suspected role in growth, metamorphosis, skeletogenesis
*Protostomia*	*Lophotrochozoa*	*Annelida*	Ligand-activated by T3 or TRIAC	Regulator of developmental transition from trochophore to crawling larva
		*Mollusca*	Presumably ligand-activated and/or controlled by alternative signaling pathways	Suspected role in growth and developmental transitions
		*Platyhelminthes*	Presumably ligand-activated and/or controlled by alternative signaling pathways	Suspected role in growth
*Non-Bilateria*	*Radiata*	*Cnidaria*	Absent from the genome	THs with a role in metamorphosis, strobilation, skeletogenesis

**Table 3 genes-12-00083-t003:** Summary of retinoic acid receptor (RAR) functions in marine invertebrates.

Taxon	Clade	Phylum	Receptor Activity	Developmental Function
*Deuterostomia*	*Chordata*	*Tunicata*	Ligand-activated by retinoic acid	Neurogenesis, budding
		*Cephalochordata*	Ligand-activated by retinoic acid	Neurogenesis, axial patterning
	*Ambulacraria*	*Echinodermata*	Ligand-activated by high concentrations of retinoic acid	Presumably involved in developmental growth
*Protostomia*	*Ecdysozoa*	*Priapulida*	Ligand-activated by high concentrations of retinoic acid	Unknown
		*Hexapoda*	Absent from the genome	RA with role in nervous system regeneration, tissue repair
	*Lophotrochozoa*	*Annelida*	Ligand-activated by high concentrations of retinoic acid	Neurogenesis
		*Mollusca*	Ligand-binding pocket occluded and potential activation by liganded RXR	Neurogenesis
*Non-Bilateria*	*Radiata*	*Cnidaria*	Absent from the genome	RA with role in neurogenesis, metamorphosis, strobilation

**Table 4 genes-12-00083-t004:** Summary of estrogen receptor (ER) and estrogen-related receptor (ERR) functions in marine invertebrates.

Taxon	Clade	Phylum	Receptor Activity	Developmental Function
*Deuterostomia*	*Chordata*	*Tunicata*	ERR: Orphan receptor ER: Lost	ERR: Suggested role in sensory cell differentiation in the larval brain
		*Cephalochordata*	ERR: Orphan receptor ER: Unknown	ERR: Suspected role in establishment of neuromuscular contacts ER: Unknown
*Protostomia*	*Ecdysozoa*	*Arthropoda*	ERR: Orphan receptor ER: Lost	ERR: Control of metabolism underlying larval growth and cell proliferation
	*Lophotrochozoa*	*Annelida*	ERR: Unknown ER: Ligand-activated receptor binding estrogens	ER: Regulation of formation and proliferation of primordial germ cells
		*Mollusca*	ERR: Orphan receptor ER: Occluded ligand binding pocket, but constitutive transcriptional activity	ERR: Unknown ER: Unknown
		*Rotifera*	ERR: Unknown ER: Ligand-activated receptor binding estrogens	ERR: Unknown ER: Unknown
*Non-Bilateria*	*Radiata*	*Cnidaria*	ER-like: Ligand-activated receptor binding paraestrol A, an ancestral estrogen	ER-like: Unknown

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
