# Peer review of "Nuclear Receptors and Development of Marine Invertebrates"

_genes, 2021, doi:10.3390/genes12010083_

Round 1

Reviewer 1 Report

SUMMARY
=======

In their manuscript entitled "Nuclear receptors and development of marine invertebrates" (genes-10566051), Miglioli et al. reviewed the current knowledge about the evolution of nuclear receptors (NR) in animals with a particular focus on marine invertebrate. They described the basic structure of NRs and their evolution in metazoa. They then detailed the NRs involved in marine invertebrate development as well as their diversification, always balancing their point of view with what is known in vertebrates for which the role and diversity of NRs have been extensively studied. The authors then focused on 4 major families of subfamilies-group of NR: THR, RAR, RXR and ER-ERR. They finally concluded their review on the potential influence of endocrine disrupting chemicals on development of marine invertebrate through the possible interaction with NRs.

The authors described thoroughly the different aspects of the nuclear receptors and the information seemed relevant. The manuscript globally reads well and is accessible for non specialists of the field. I think this review can be relevant for the field. However, I have some comments that I hope will help improving the review. This should be easily addressed.

LEGEND
======
* l.: line
* Fig.: figure
* Tab.: table

This review is written in markdown (see https://en.wikipedia.org/wiki/Markdown for more information).

COMMENTS
========

* Length: While the manuscript reads well, it is quite lengthy (15 pages of single spaced lines). The topic is large but I encourage the authors to trim the text when possible. Otherwise readers may be discouraged to read the review.

* Acronyms: There is a lot of acronyms used throughout the manuscript. They are not necessarily defined when first encountered. While they are available in table 1, this slightly degrades the reading experience. I would encourage the authors to pay a particular attention to this.

* Relevance of the families reviewed: The authors reviewed information related to THR, RAR, RXR and ER-ERR families and introduced their paragraphs on why these families are important usually in vertebrates. But it is unclear why they discussed these families specifically. Were they specifically relevant for invertebrates? And if yes, why?

Comments on the text
--------------------

* **l. 313**: "[[89,90]]" should be "[89,90]".

Comments on the figures
-----------------------

* **Fig. 1**: The box around the structure is not needed.

* **Fig. 2**: What does the "#" mean? I suspect this is the same meaning as in table 1 but this is not stated in the caption.

* **Fig. 2, l. 62**: "Arrowheads": arrowheads are probably not the best representation. In evolution, this usually means expansion of an element. I would suggest finding an alternative to the arrowheads to symbolize appearance.

* **Fig. 3**: The acronyms in red and green were not detailed in the figure caption. I would refer to table 1.

Comments on the tables
----------------------

* **Tab. 1, l. 120**: "shown in red": There was no red in the table. Maybe this is not allowed by the journal. Bold could be an alternative.

* **Tab. 1, l. 126**: "The table is divided in part 1 (NR#-1), 2 (NR2-4) and 3 (NR5-7/8).": I think this is not needed as the table follows a logical order.

Reviewer 2 Report

The manuscript is entitled Nuclear receptors and development of marine invertebrates described the finding of NRs and correlations in marine invertebrates' development, which was written comprehensively. The manuscript is quite informative and readable by the authors' thorough understanding.

Author Response

Answer to reviewer 2 :

The manuscript is entitled Nuclear receptors and development of marine invertebrates described the finding of NRs and correlations in marine invertebrates' development, which was written comprehensively. The manuscript is quite informative and readable by the authors' thorough understanding.

Thank you very much.

Reviewer 3 Report

I have opportunity to read the MS entitled "Nuclear receptors and development of marine invertebrates" by Miglioli.  The MS is well organized on a wide range.  The MS will be interested for the readers in this field.

I have a concern in Table 1.  In columns of “Physiological Ligand”, please explain the means of blanks in the table legends. 

Author Response

Answer to reviewer 3:

I have opportunity to read the MS entitled "Nuclear receptors and development of marine invertebrates" by Miglioli.  The MS is well organized on a wide range.  The MS will be interested for the readers in this field.

I have a concern in Table 1.  In columns of “Physiological Ligand”, please explain the means of blanks in the table legends.

In the table description the authors stated as follows:

Line 57: “When information is absent or currently under discussion cells are left empty”

Accordingly, the authors left blank spaces non only in the ligand column but also in other columns/rows of the table.

The term Physiological ligand refers to the active primary ligand activating the receptor in physiologic and in vivo conditions. Moreover, the fact that a physiological ligand is yet to be identified does not necessarily mean that the NR is an orphan receptor. As the goal of the manuscript is to report extant information, the authors avoided any speculation regarding the orphan and/or liganded state of NRs in the table unless the topic was discussed in the text with references in support.

In order to be clearer, the authors modified the text as follows:

Line 57: “When information is absent or currently under discussion cells are left empty”

In addition we have noticed a mistake in Fig 3C (Line 89): Ambulacra has been changed to ambulacraria

We are hoping that this new version of the manuscript will be accepted for publication.

Sincerely.